# Expression profile of sonic hedgehog signaling-related molecules in basal cell carcinoma

Hye Sung Kim[1]☯, Young Sil Kim[1]☯, Chul Lee[2], Myung Soo Shin[3], Jae Wang Kim[4], Bo Gun Jang[1]*

**1** Department of Pathology, Jeju National University College of medicine, Jeju, South Korea, **2** Department of Pathology, Seoul National University College of medicine, Seoul, South Korea, **3** Department of Plastic Surgery, Jeju National University College of medicine, Jeju, South Korea, **4** Department of Dermatology, Jeju National University College of medicine, Jeju, South Korea

☯ These authors contributed equally to this work.
* Bgjang9633@gmail.com

**Data Availability Statement:** All relevant data are within the paper.

**Funding:** This study was supported by a research grant from Jeju National University Hospital in

## Abstract

Basal cell carcinoma (BCC) is the most common human cancer, characterized by aberrant activation of the hedgehog (HH) signaling pathway resulting from mutations in the patched 1 (*PTCH1*) or smoothened (*SMO*) genes. In the present study, to uncover the expression profile of HH signaling-related molecules, we thoroughly examined the mRNA and protein expression levels of six molecules including GLI1, GLI2, PTCH1, PTCH2, SHH, and SMO in BCC and various other cutaneous tumors. Real-time PCR analysis demonstrated that BCC showed remarkably enhanced mRNA expression of all HH molecules, except SMO compared to other skin tumors. However, immunohistochemical analysis revealed that only GLI1 protein was specifically upregulated in BCC, while the other HH-related proteins did not show any significant differences between the tumors. Notably, other skin malignancies such as squamous cell carcinoma, sebaceous carcinoma, and malignant melanoma showed no GLI1 expression and there was no difference in GLI1 expression between the BCC subtypes. In addition, GLI1 and GLI2 expression were strongly associated with the hair follicle stem cell markers, LGR4 and LGR5, which are known target genes of the Wnt pathway. Our results suggest that GLI1 has the potential to be a diagnostically useful marker for differentiating BCC from other skin malignancies and an interaction between the HH and Wnt signaling pathways may be involved in the development of BCCs.

## Introduction

Basal cell carcinoma (BCC) is the most common human cancer, and approximately 750, 000 BCCs are diagnosed each year in the United States alone [1]. BCCs are largely caused by exposure to ultraviolet light and develop on the sun-exposed areas of skin. They are classically slow-growing and locally invasive cancers that are considered to arise from hair follicles [2, 3]. The majority of BCCs occur sporadically, however, basal cell nevus syndrome (BCNS, also

2017. The funders had no role in study design, data collection and analysis, decision to publish, or preparation of the manuscript.

**Competing interests:** The authors have declared that no competing interests exist.

known as Gorlin syndrome) is a rare heritable disease, in which the patients have a marked susceptibility to developing BCCs. Family-based linkage studies identified patched 1 (*PTCH1*) as the causative mutant gene, indicating that aberrant hedgehog (HH) signaling activity is responsible for the development of BCCs [4, 5]. Since the discovery of *PTCH1* mutation, it has been demonstrated that most spontaneous BCCs are associated with mutations in the components of the HH signaling pathway such as *PTCH1*, Smoothened (*SMO*), and suppressor of fused homolog (*SUFU*) leading to constitutive activation of the HH signaling [1, 5, 6]. In addition, the growing body of evidence suggests that dysregulation of the HH signaling pathway occurred frequently in a wide variety of cancers [7].

In the canonical HH pathway, sonic hedgehog (SHH) functions as an initiator and PTCH1 is a 12-transmembrane receptor for SHH that has a regulatory effect on the pathway [8]. In the absence of SHH, PTCH1 binds to SMO and inhibits the downstream signaling cascade, whereas the binding of SHH to PTCH1 relieves SMO inhibition, resulting in activation of the downstream zinc-finger glioma transcription factor (GLI) family of transcription factors, GLI1, GLI2, and GLI3 [1]. GLI1 appears to exclusively act as a transcriptional activator, whereas GLI2 and GLI3 can display both activator and repressor functions [9]. In the absence of upstream signal, GLI3, and to a lesser degree GLI2, are proteolytically cleaved and play a role of transcriptional repressors [10]. The nuclear localization of GLI1 is considered to be characteristic of the activated HH signaling pathway [11]. Although several reports have specifically shown GLI1 expression in human BCCs [12–14], no study has thoroughly examined the expression of multiple HH-related molecules in a variety of human skin neoplasms. In this study, we investigated the expression profile of six HH pathway molecules (GLI1, GLI2, PTCH1, PTCH2, SMO, and SHH) in BCCs and other benign and malignant skin tumors by real-time PCR and immunohistochemistry. Although the precise cellular origin of BCC has been controversial, recent studies have demonstrated that BCC-like tumors can arise from multiple hair follicle (HF) stem cell populations [15–17]. Since several distinct stem cell markers of HF have been identified by lineage-tracing experiments, we also assessed the correlation of HH molecules with the established markers including *LGR4*, *LGR5*, *LGR6*, and *LRIG1*.

## Materials and methods

### Subjects

A total of 152 formalin-fixed, paraffin-embedded (FFPE) human skin tissues (normal skin, n = 5; skin tumors, n = 168) were collected from punch or excisional biopsy specimens at the Jeju National University Hospital, Jeju from 2011 to 2015. Skin tumors are basal cell carcinomas (BCCs; n = 84), trichoepitheliomas (TEs; n = 17), pilomatricomas (PMCs; n = 13), eccrine poromas (EPs; n = 11), spiradenomas (SPAs; n = 9), hidradenomas (HDAs; n = 10), squamous cell carcinomas (SCCs; n = 9), sebaceous carcinomas (SBCs; n = 7), and melanomas (MNs; n = 8). All hematoxylin and eosin-stained slides were thoroughly re-examined by two dermatopathologists (BGJ and CL), and only clear cases with typical histologic features were included for the study. Additionally, surgically resected specimens (18 BCCs, 7 TEs, 8 PMCs, 9 EPs, 8 SPAs, 10 HDAs, 9 SCCs, 7 SBCs, 8 MNs, and 5 normal skin tissues) were chosen for RNA extraction and real-time PCR analysis. This study was approved by the Institutional Review Board of Jeju National University Hospital (2017-06-005). Informed consent was waved due to the retrospective nature of this study and all the data were analyzed anonymously.

### Tissue microarray construction

We constructed ten tissue microarrays (TMAs) containing 84 BCCs, 17 TEs, 13 PMCs, 11 EPs, 9 SPAs, 10 HDAs, 9 SCCs, 7 SBCs, 8 MNs and 5 normal skins. In brief, a representative tumor

area (4 mm in diameter) was extracted from each FFPE tumor tissue (donor blocks) and arranged in a new recipient paraffin block (tissue microarray block) using a trephine apparatus (SuperBioChips Laboratories, Seoul, Korea).

## Immunohistochemistry

Immunohistochemistry (IHC) was performed on 4-μm TMA sections using a Ventana Bench-Mark XT Staining systems (Leica Microsystems, Wetzlar, Germany) according to the manufacturer's instructions. The primary antibodies used are as follows: anti-GLI1 (Cell signaling, #3538), anti-GLI2 (Abcam, ab26056), anti-PTCH1 (Abcam, ab53715), anti-PTCH2 (Abcam, ab238338), anti-SHH (Abcam, ab53281), and anti-SMO (Abcam, ab236465). GLI1 and GLI2 were evaluated for cytoplasmic and nuclear stain, while PTCH1, PTCH2, SHH, and SMO were evaluated for cytoplasmic stain. IHC was scored from 0 to 3 according to the stain intensity because a majority of cases showed a diffuse staining pattern.

## RNA extraction and quantitative real-time PCR

Each tumor area was manually dissected from FFPE tissue section (4-μm thick) from a representative paraffin block. Total RNA was extracted with an RNeasy FFPE Kit (Qiagen, Valencia, CA, USA) with a slight modification as previously described [18]. The cDNA was synthesized from 1–2 μg of RNA with random hexamer primers using the GoScript reverse transcription system (Promega, Madison, Wisconsin, USA). Real-time PCR was performed with a StepOne Plus real-time PCR system (Applied Biosystems, Foster City, CA, USA) using the Premix Ex Taq (Takara Bio, Shiga, Japan) according to the manufacturer's instructions. The cycling conditions are as follows: initial denaturation for 30 s at 95˚C, followed by 40 cycles of 95˚C for 1 s and 60˚C for 5 s. The following TaqMan gene expression assays were used: Hs01551772_m1 (*LGR4*), Hs00173664_m1 (*LGR5*), Hs00663887_m1 (*LGR6*), Hs01006146_m1 (*LRIG1*), Hs00171790_m1 (*GLI1*), Hs01119974_m1 (*GLI2*), Hs00181117_m1 (*PTCH1*), Hs00184804_m1 (*PTCH2*), Hs01123832_m1 (*SHH*), Hs01090242_m1 (*SMO*) and Hs0275899_g1 (*GAPDH*). *GAPDH* served as the endogenous control.

## RNA in situ hybridization

In situ hybridization (ISH) for *LGR5* was performed using the RNAscope FFPE assay kit (Advanced Cell Diagnostics, Inc., Hayward, CA, USA) as described previously [18]. In brief, 4-μm thick FFPE tissue sections were baked at 60˚C for 1 hour, followed by protease digestion, and subject to hybridization with *LGR5* for 2 hours. An HRP-based signal amplification system was hybridized to the probe before color development with 3,3′-diaminobenzidine tetrahydrochloride. The housekeeping gene, *ubiquitin C* and the bacterial gene, *DapB* served as a positive and negative controls, respectively. Brown punctate dots in the nucleus and/or cytoplasm were considered positive.

## Statistical analysis

Statistical analyses were performed with Prism version 5.0 (GraphPad Software, Inc., San Diego, CA, USA). Comparisons between the groups of real-time PCR data were tested using Tukey's multiple comparison test. The significance of the correlations between *LGR 4*, *LGR5*, *LGR6* and HH signaling-related genes was assessed with the Pearson correlation test. The correlations between IHC score and BCC subtypes were tested by Pearson chi-square test. A *P*-value < 0.05 was considered statistically significant.

## Results

### Real-time PCR analysis for hedgehog signaling-related genes in various skin tumors

To measure the transcription levels of HH-related molecules in human skin tumors, the FFPE samples were collected as follows: normal skin (n = 5), BCC (n = 18), TE (n = 7), PMC (n = 8), EP (n = 9), HDA (n = 10), SPA (n = 8), SCC (n = 9), SBC (n = 7), and MN (n = 8). Real-time PCR results demonstrated that all HH-related molecules, except *SMO*, were remarkably elevated in BCC compared to normal skin and most other skin tumors (Fig 1). TE also showed higher expression of *GLI1*, *GLI2*, and *SHH* than normal skin, however, the expression levels of *PTCH1* and *PTCH2* were as low as normal skin tissues (Fig 1C and 1D). *SHH* expression was slightly higher in several tumors including BCC, TE, EP, and SPA (Fig 1E). Interestingly, *SMO* levels were

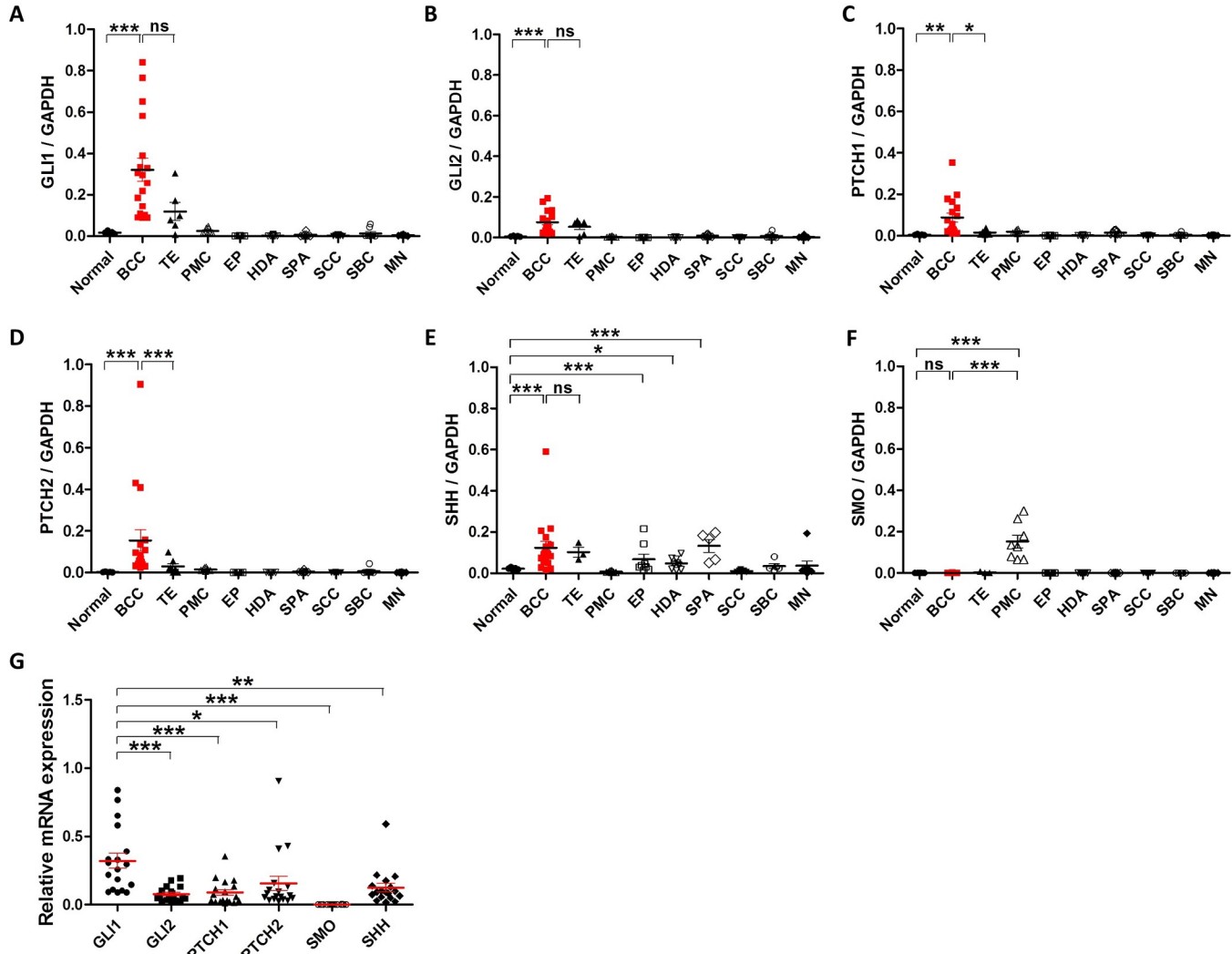

**Fig 1. Relative mRNA levels of Hedgehog signaling-related molecules in various skin tumors.** Expression of GLI1 (A), GLI2 (B), PTCH1 (C), PTCH2 (D), SHH (E), and SMO (F) in normal skin tissue (n = 5), basal cell carcinoma (BCC; n = 18), trichoepithelioma (TE; n = 7), pilomatricoma (PMC; n = 8), eccrine poroma (EP; n = 9), hidradenoma (HAD; n = 10), spiradenoma (SPA; n = 8), squamous cell carcinoma (SCC; n = 9), sebaceous carcinoma (SBC; n = 7), and malignant melanoma (MN; n = 8). (G) Relative expression of Hedgehog signaling molecules in BCCs (n = 18). Bars represent mean ± SEM. *P < 0.05; **P < 0.01; ***P < 0.001; ns, not significant.

significantly increased only in PMC ([Fig 1F]). When comparing each HH-related molecule in BCC, GLI1 was the most highly expressed, while *SMO* was the least expressed ([Fig 1G]).

## Immunohistochemical analysis of Hedgehog signaling-related molecules in skin tumors

Next, we performed IHC to examine the protein expression of HH-related molecules using tissue microarrays from a variety of skin tumors as follows: BCC (n = 84), TE (n = 17), PMC (n = 13), EP (n = 11), HDA (n = 10), SPA (n = 9), SCC (n = 19), SBC (n = 9), and MN (n = 8). The mean IHC scores demonstrated that GLI1 exhibited the most differential expression between skin tumors ([Fig 2A]), whereas the GLI2, PTCH1, PTCH2, SHH, and SMO proteins showed less or no significant difference in expression ([Fig 2B–2F]). GLI1 is specifically expressed in the bulb areas of hair follicles ([Fig 3A]), but not in the sweat glands ([Fig 3B]) or sebaceous glands ([Fig 3C]). Most BCCs expressed GLI protein, which tended to be notably stronger in the palisading cells of the tumor nests ([Fig 3D]). TEs also showed GLI expression but not as much as BCCs ([Fig 3E]). Tumors other than BCC and TE did not express GLI1 ([Fig 3F–3L]).

## Expression profile of hedgehog signaling-related molecules in basal cell carcinoma

In BCCs, GLI 1 was most highly expressed among the HH-related proteins, which was consistent with the mRNA expression levels ([Fig 4A]). Representative images are shown in [Fig 4B]. Since BCCs are classified into four subtypes: nodular, micronodular, superficial, and infiltrative (or desmoplastic), we examined whether there is a difference in GLI1 expression

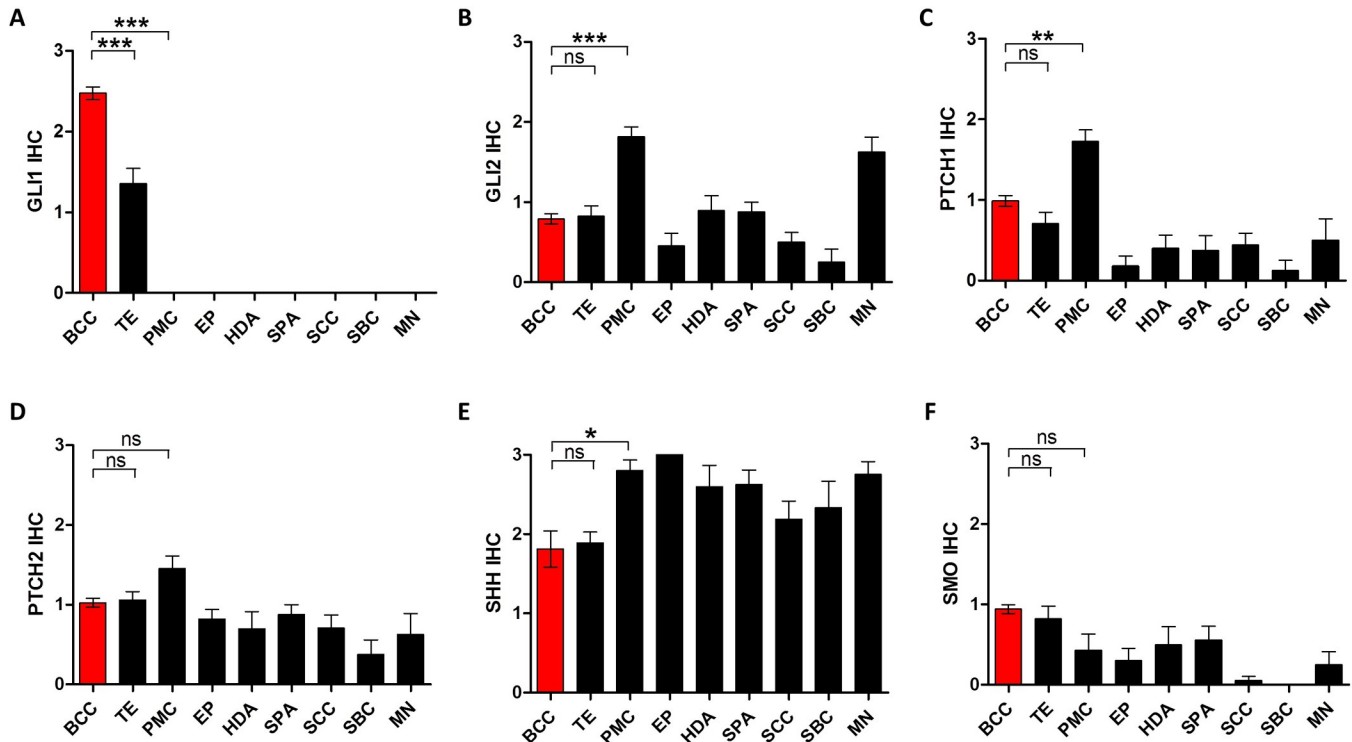

**Fig 2. Immunohistochemical analysis of hedgehog-signaling related molecules in various skin tumors.** Expression of GLI1 (A), GLI2 (B), PTCH1 (C), PTCH2 (D), SHH (E), and SMO (F) in basal cell carcinoma (BCC; n = 84), trichoepithelioma (TE; n = 17), pilomatricoma (PMC; n = 13), eccrine poroma (EP; n = 11), hidradenoma (HAD; n = 10), spiradenoma (SPA; n = 9), squamous cell carcinoma (SCC; n = 19), sebaceous carcinoma (SBC; n = 9), and malignant melanoma (MN; n = 8). Bars represent mean ± SEM. *P < 0.05; **P < 0.01; ***P < 0.001; ns, not significant.

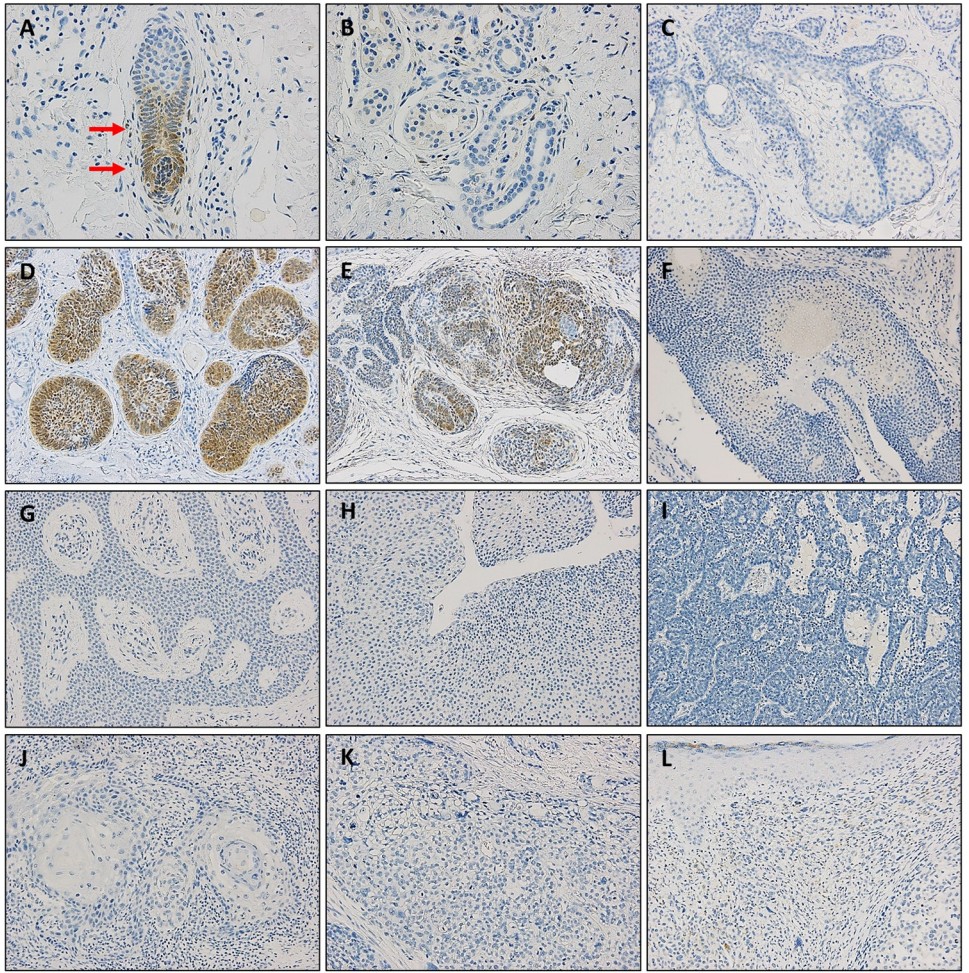

**Fig 3. Immunohistochemical staining for GLI1 in normal skin and various skin tumors.** Representative images of GLI expression in hair follicle (A), sweat gland (B), sebaceous gland (C), basal cell carcinoma (D), trichoepithelioma (E), pilomatricoma (F), eccrine poroma (G), hidradenoma (H), spiradenoma (I), squamous cell carcinoma (J), sebaceous carcinoma (K), and malignant melanoma (L). A, B: ×400 magnification, C-L: ×200 magnification.

according to subtypes. However, all subtypes of BCCs showed similar levels of HH-related molecules including GLI1 (Fig 5).

## Correlation of GLI1 expression with epidermal stem cell markers

Several molecules including LGR4, LGR5, LGR6, Keratin 15, Sox9, CD34, Blimp1 and LRIG1 have been identified as stem cell markers of the epidermis and hair follicles [19] and HH signaling is involved in the regulation of discrete populations of stem and progenitor cells in various organs including skin [20]. Thus, we explored the correlation of HH-related molecules except *SMO* with some of the stem cell markers; *LGR4, LGR5, LGR6*, and *LRIG1*. Interestingly, *GLI1* expression showed a strong positive correlation with *LGR4* ($r^2 = 0.62$, $P < 0.001$), *LGR5* ($r^2 = 0.60$, $P < 0.001$), but not with *LGR6* ($r^2 = 0.21$, $P = 0.05$) and *LRIG1* ($r^2 = 0.07$, $P = 0.27$) (Fig 6A). Representative images of a BCC expressing both *LGR5* mRNA and GLI1 protein are shown in Fig 6B. *GLI2* and *PTCH1* also had a positive correlation with *LGR4* and *LGR5*, while *PTCH2* and *SHH* showed a correlation only with *LGR5* (Fig 7).

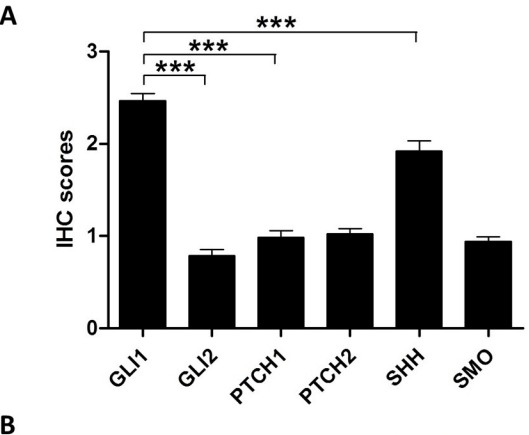

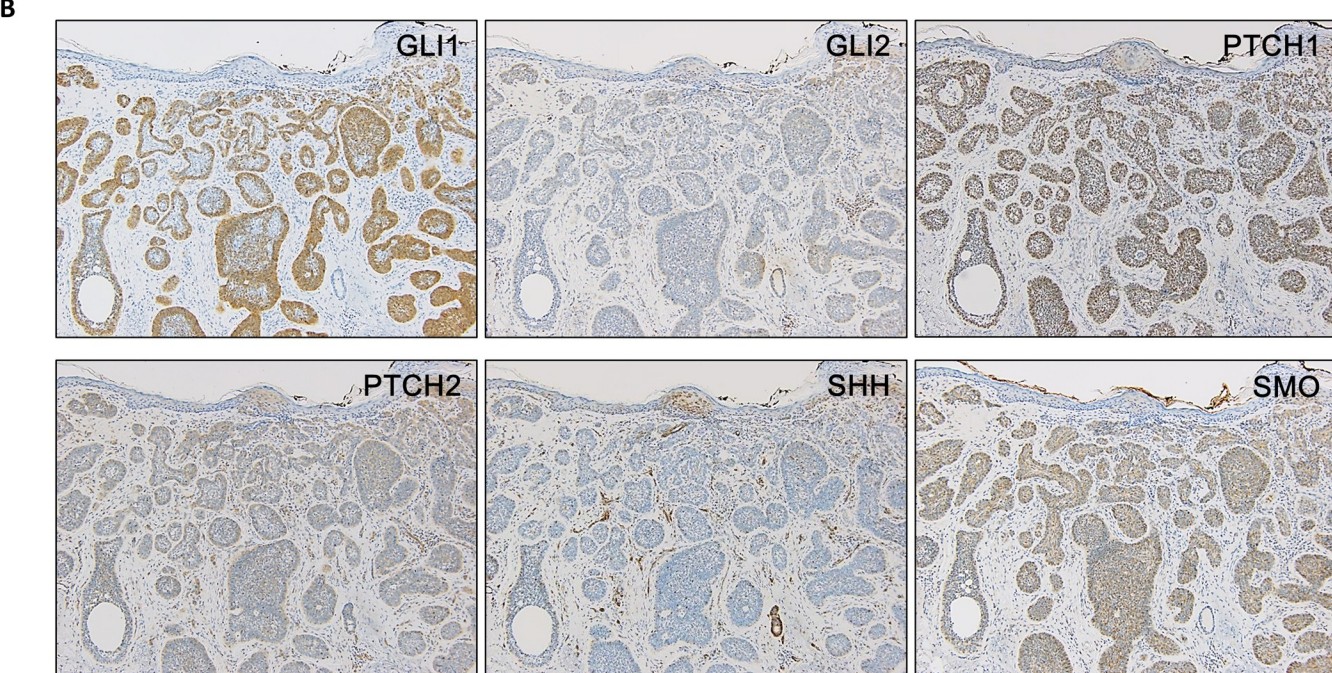

**Fig 4. Expression of hedgehog (HH)-signaling related molecules in basal cell carcinomas.** (A) A graph showing the immunohistochemistry scores of HH-related proteins in basal cell carcinoma (n = 81). (B) Representative images of H&E and immunohistochemical staining for GLI1, GLI2, PTCH1, PTCH2, SHH, and SMO in a basal cell carcinoma. B: ×100 magnification. ***P < 0.001.

## Discussion

In this study, we first measured the mRNA levels of six HH pathway molecules to investigate the transcriptional activity of the HH signaling pathway in various cutaneous tumors. As expected, out of the nine skin tumors examined, only BCCs exhibited substantially elevated levels of all the HH-related molecules, except SMO. This finding is consistent with previous studies reporting that HH signaling can activate genes involved in positive and negative feedback such as *GLI1*, *GLI2*, *PTCH1*, and *PTCH2* [21–23]. Although TEs also expressed slightly increased levels of *GLI1*, *GLI2*, *PTCH2*, and *SHH*, it was significantly lower than BCCs. Interestingly, *SMO* transcription was not altered in BCCs, whereas Martinez *et al.* recently showed an increased mRNA expression of *SMO* in BCCs from 20 nevoid basal cell carcinoma syndrome (NBCCS) patients mainly caused by *PTCH1* gene mutations [24]. This discrepancy is probably due to the different study groups because our study only includes sporadic BCCs that

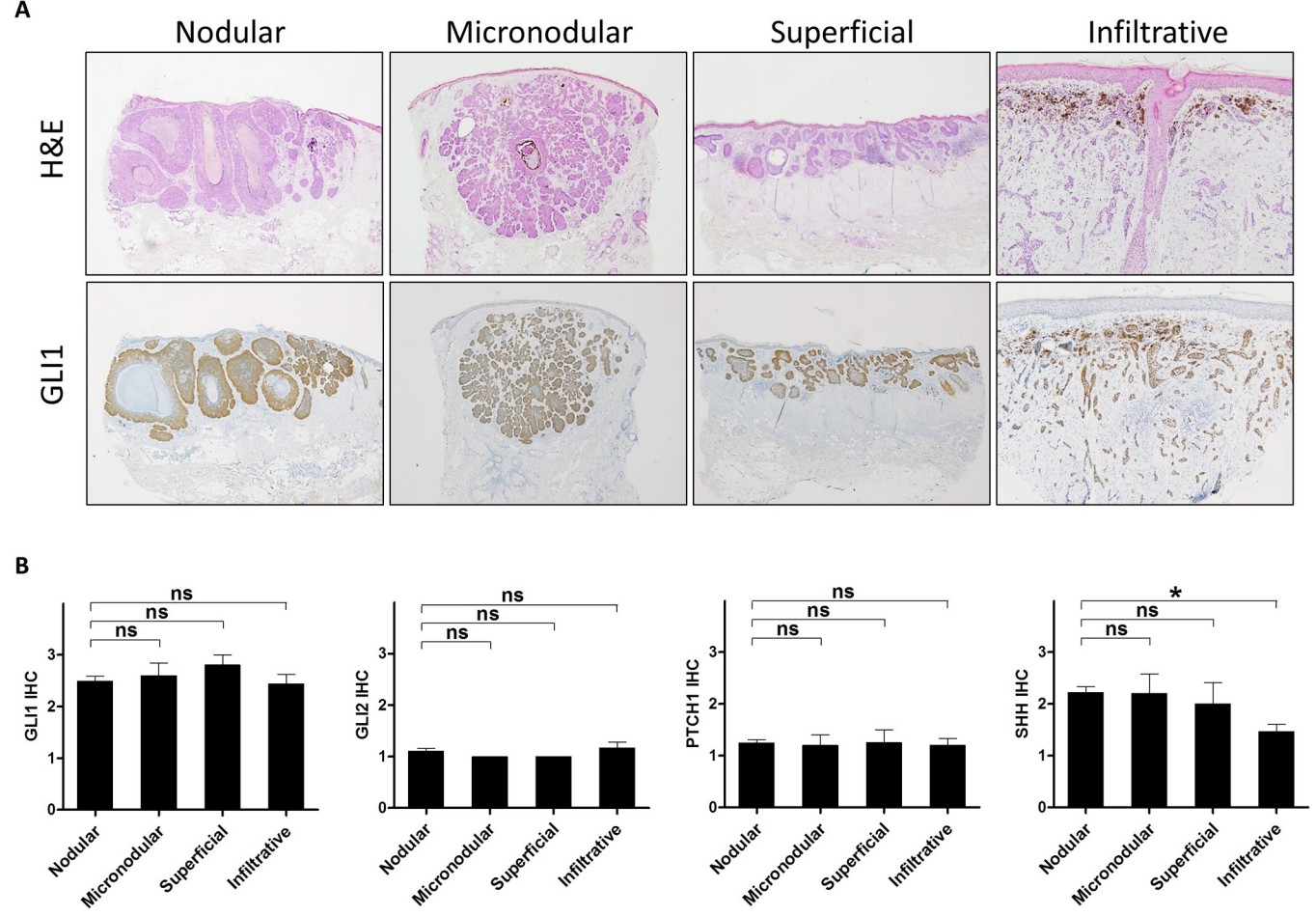

**Fig 5. GLI1 expression in each subtype of basal cell carcinoma.** (A) Representative H&E and GLI1 images of nodular, micronodular, superficial and infiltrative basal cell carcinoma. Nodular, micronodular, superficial and infiltrative basal cell carcinomas: ×40 magnification, Infiltrative: ×100 magnification. (B) No differences were observed in the expression of GLI1, GLI2, PTCH1, and SHH between each subtype of basal cell carcinoma including nodular (n = 55), micronodular (n = 5), superficial (n = 5) and infiltrative (n = 16) types.

have different genetic profiles from those of NBCCS patients even though both BCCs share a common signaling abnormality.

Immunohistochemical evaluation of the HH-related molecules demonstrated that only GLI1 protein had the same expression pattern as its mRNA in skin tumors. Real-time PCR analysis showed that *GLI2*, *PTCH1*, and *PTCH2* expression were specifically elevated in BCCs and TEs, however, IHC analysis showed that there were no significant differences in their protein expression levels between the skin tumors. This finding was not surprising since mRNA and protein expression is often discordant due to several biological reasons, such as post-transcriptional modification and different degradation rates. For example, the lower GLI2 protein expression may be in part associated with the high turnover rate considering the fact that GLI2 can be proteolytically processed into the truncated-repressor form in the absence of HH ligands. Additionally, the discrepancy between mRNA and protein levels could be in part due to the low sensitivity and specificity of antibodies used in the present study. Indeed, only GLI1 antibody, C68H3, was previously demonstrated to be reliable for IHC [14], and was further confirmed in our studies by GLI1 expression observed in normal skin tissues, in which C68H3 antibody exhibited exclusive and specific expression in hair follicles (Fig 3).

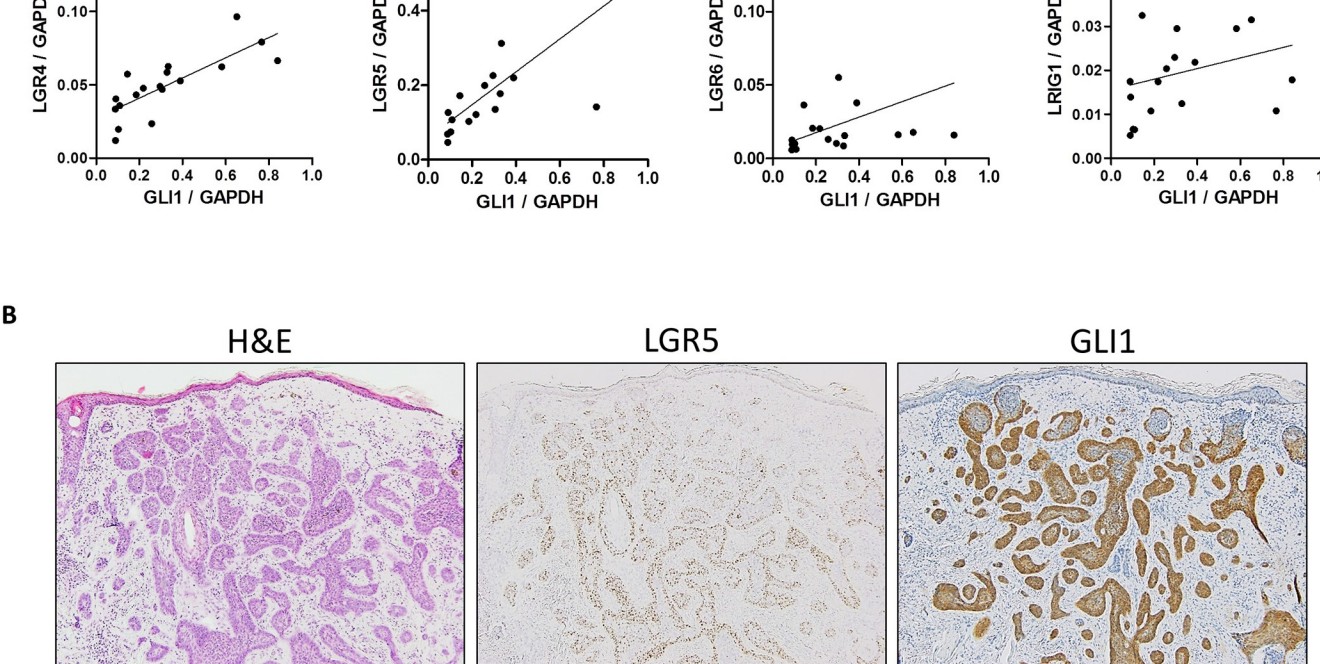

**Fig 6. Correlations of *GLI1* expression with other stem cell markers in basal cell carcinomas.** (A) Scatter plots with regression lines showing the correlations between *GLI1* and stem cell marker mRNA expression in basal cell carcinomas (n = 18). (B) Representative case of basal cell carcinoma expressing both *LGR5* mRNA and GLI1 protein (RNA in situ hybridization for *LGR5* and immunohistochemistry for GLI1). B: ×100 magnification.

GLI1 expression was highest among the HH-related molecules in BCCs at both mRNA and protein levels. In addition, GLI1 was only detected in BCCs and TEs, whereas it was not observed in other benign and malignant skin tumors. It has been well-established that BCCs and TEs, which share similar histologic features, have upregulated GLI1 expression due to aberrant HH signaling, which plays a decisive role in the development of both tumors [25]. In addition, it was notable that only BCCs were found to express GLI among the malignant skin tumors, suggesting GLI1 as a candidate diagnostic marker in differentiating BCC from other skin malignancies. In particular, GLI1 may be useful to differentiate BCCs from basaloid squamous cell carcinoma of the skin, another basaloid tumor that is rare but diagnostically challenging.

Multiple stem cell markers have been identified in the mouse skin and hair follicles including LGR4, LGR5, LGR6, LRIG1, SOX9 and Keratin 15 [26–28]. We previously demonstrated that *LGR5* and *LGR6* expression was upregulated in BCCs [29], and here we examined whether HH signaling-related molecules were associated with those stem cell markers. Interestingly, *GLI1* and *GLI2* have strong correlations with stem cell markers, particularly *LGR4* and *LGR5* (Figs 6 and 7). This finding seems to be consistent with a previous study reporting that GLI1-expressing stem cells co-express LGR5 [30]. In this study, we showed that GLI1-expressing cells reside in the hair bulb area, which largely overlapped with the *LGR5*-expressing area. Thus, it is reasonable to hypothesize that the cells expressing *LGR5*, as well as GLI1, in the bulb area might be the origin of cells giving rise to BCC. Since *LGR4* and *LGR5* are the target genes of the Wnt signaling pathway, our results also suggest a possible interplay between the HH and

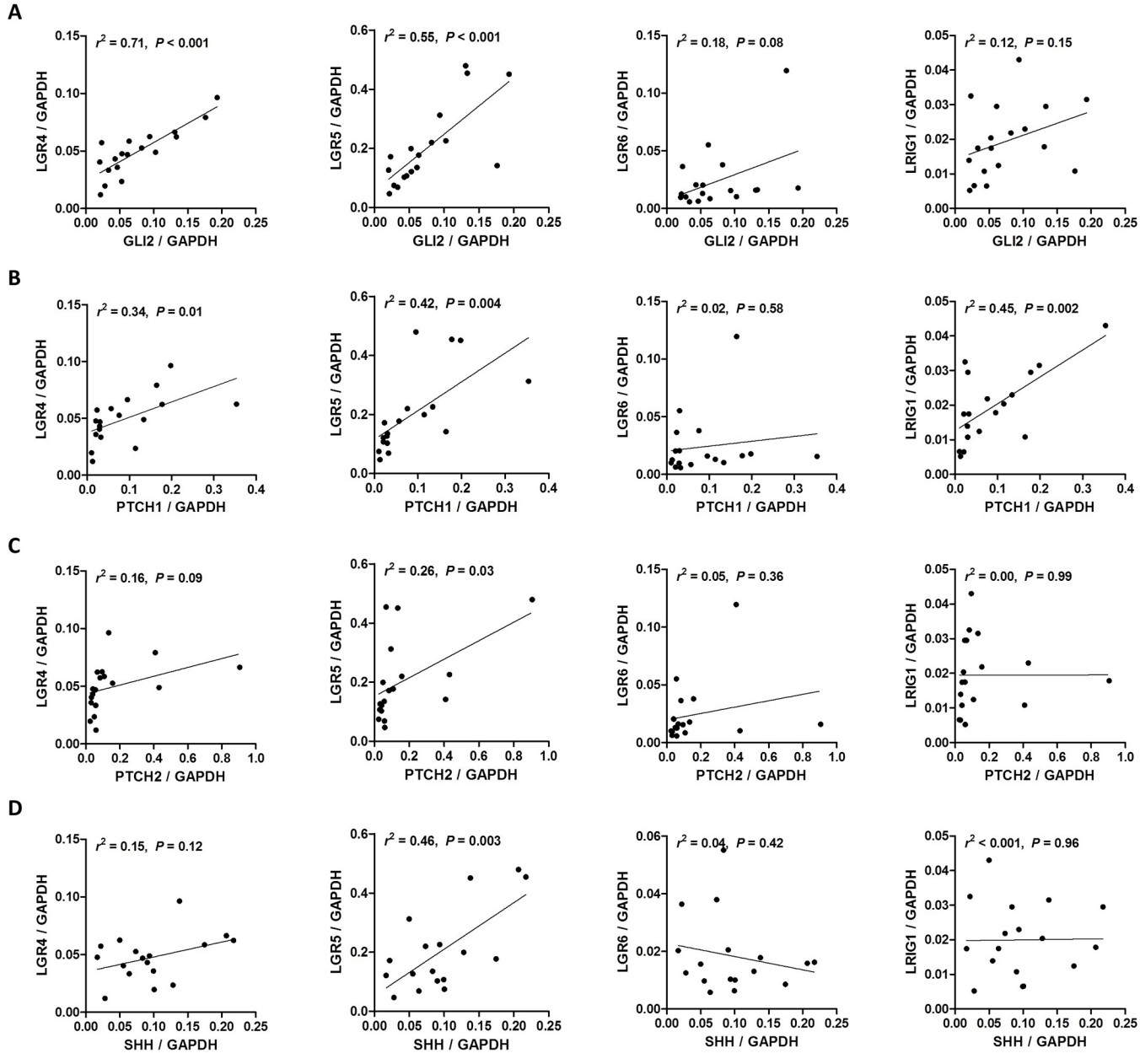

**Fig 7. Correlation of *GLI2*, *PTCH1*, *PTCH2*, and *SHH* with epidermal stem cell markers in basal cell carcinoma.** Scatter plots with regression lines showing the correlations of *GLI2* (A), *PTCH1* (B), *PTCH2* (C), and *SHH* (D) with stem cell markers including *LGR4*, *LGR5*, *LGR6*, and *LRIG1*.

Wnt signaling pathways. It has been demonstrated that the Wnt-HH signaling axis is essential for maintaining the hair cycle and activation of canonical Wnt signaling induces SHH expression [31].

In summary, we investigated the mRNA and protein expression profile of six HH signaling-related molecules in a variety of cutaneous benign and malignant neoplasms. Our results demonstrated that BCC shows dramatically increased mRNA levels of *GLI1*, *GLI2*, *PTCH1*, *PTCH2*, and *SHH*. Among these 5 molecules, only GLI1 showed elevated protein expression in BCCs, suggesting a possible role as a diagnostic marker for differentiating BCCs from other

skin malignancies. Furthermore, *GLI1* was strongly associated with hair follicle stem cell markers, *LGR4* and *LGR5*, suggesting a link between HH and Wnt signaling in BCC carcinogenesis.

## Acknowledgments

We would like to thank Hye Jung Lee and Seung Hi Jeong for their technical support.

## Author Contributions

**Conceptualization:** Bo Gun Jang.

**Data curation:** Chul Lee, Myung Soo Shin, Jae Wang Kim.

**Funding acquisition:** Young Sil Kim.

**Investigation:** Hye Sung Kim, Young Sil Kim, Bo Gun Jang.

**Methodology:** Young Sil Kim.

**Resources:** Chul Lee.

**Validation:** Jae Wang Kim.

**Visualization:** Hye Sung Kim, Young Sil Kim.

**Writing – original draft:** Hye Sung Kim.

**Writing – review & editing:** Bo Gun Jang.

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
