## [Decision Letter · Decision Letter 0]

9 Oct 2019

PONE-D-19-24006

Expression profile of Sonic hedgehog signaling-related molecules in basal cell carcinoma

PLOS ONE

Dear Dr Jang,

Thank you for submitting your manuscript to PLOS ONE. After careful consideration, we feel that it has merit but does not fully meet PLOS ONE’s publication criteria as it currently stands. Therefore, we invite you to submit a revised version of the manuscript that addresses the points raised during the review process.  Many of the points to be addressed are in the attached comments.

1.  The authors should include in their interpretation of results and in the discussion consideration that GLI1 and GLI2 can be full-length or cleaved which impacts their function.  Details on where in the protein antibodies bind (and implications for protein function) should be included.  (A reminder that full length westerns need to be included in the supplemental materials).

2.  More discussion on how their results fit into previous findings of increased SMO levels in BCC, differences in approaches or why there may be differences should be considered. (See reviewer's comments).

3.  Address reviewer 1's queries about correlations between stem cell markers and HH-related molecules.

4.  Fix the graphs and figures per reviewer 1's suggestions.

5. In the methods and results clarify the details on number of samples that were used for each analysis. 

6. Respond to the additional points raised by the reviewer.

7. Proofread the manuscript carefully and fix the typographical errors that were identified by the reviewer.

We would appreciate receiving your revised manuscript by Nov 23 2019 11:59PM. To enhance the reproducibility of your results, we recommend that if applicable you deposit your laboratory protocols in protocols.io, where a protocol can be assigned its own identifier (DOI) such that it can be cited independently in the future. For instructions see: http://journals.plos.org/plosone/s/submission-guidelines#loc-laboratory-protocols

We look forward to receiving your revised manuscript.

Kind regards,

Amanda Ewart Toland, Ph.D.

Academic Editor

PLOS ONE

Journal Requirements:

Additional Editor Comments (if provided):

Reviewers' comments:

Reviewer's Responses to Questions

**Comments to the Author**

1. Is the manuscript technically sound, and do the data support the conclusions?

Reviewer #1: Yes

2. Has the statistical analysis been performed appropriately and rigorously? 

Reviewer #1: Yes

3. Have the authors made all data underlying the findings in their manuscript fully available?

Reviewer #1: Yes

4. Is the manuscript presented in an intelligible fashion and written in standard English?

Reviewer #1: Yes

5. Review Comments to the Author

Reviewer #1: The manuscript reveals compelling evidence about mRNA expression levels of Hedgehog pathway components (GLI1, GLI2, PTCH1, PTCH2, SHH and SMO) in different skin tumors and the expression profile observed was compared to stem cells markers (LGR4, LGR5, LGR6, and LRIG1) content. Also, protein expression and subcellular localization was studied by IHC technique. In this setting, Gli1 was found overexpressed in BCC at mRNA and protein levels. The manuscript reveal evidence that support the conclusions but some points need to be solved to be published. Please, answer the questions (provided in the attach file) in a "point by point" manner.

6. PLOS authors have the option to publish the peer review history of their article (what does this mean?). If published, this will include your full peer review and any attached files.

Reviewer #1: No

---

## [Author Response · Author response to Decision Letter 0]

18 Oct 2019

We highly appreciate the reviewer’s careful evaluation of our manuscript and constructive comments. We agree with the reviewer and tried our best to revise our manuscript in order to address all the concerns raised. 

The manuscript reveals compelling evidence about mRNA expression levels of Hedgehog pathway components (GLI1, GLI2, PTCH1, PTCH2, SHH and SMO) in different skin tumors and the expression profile observed was compared to stem cells markers (LGR4, LGR5, LGR6, and LRIG1) content. Also, protein expression and subcellular localization was studied by IHC technique. In this setting, Gli1 was found overexpressed in BCC at mRNA and protein levels. 

- In fact, some discrepancy was shown between both RNA and protein expression pattern in the other HH components. This is particularly important in GLI1 and GLI2, since both proteins can be found as activator (full-length) as well as repressor (proteolytically cleaved) forms. The authors have to take this consideration when interpret and discuss your IHC results.

: The dual role of GLI2 and GLI3 as activator and repressor is included in the Introduction (In page3, line 20-23), and its significance in IHC result has been discussed in Discussion (In page9, line 7-9).

In page3, line 20-23: “GLI1 appears to exclusively act as a transcriptional activator, whereas GLI2 and GLI3 can display both activator and repressor functions (9). In the absence of upstream signal, GLI3, and to a lesser degree GLI2, are proteolytically cleaved and play a role of transcriptional repressors (10).”

In page9, line 7-9: “For example, the lower GLI2 protein expression may be in part associated with the high turnover rate considering the fact that GLI2 can be proteolytically processed into the truncated-repressor form in the absence of HH ligands.”

Additionally, the authors not found increased SMO mRNA level in BCC, when have been recently reported increases in this messenger in BCCs from Gorlin syndrome, even after adjustment by germinal and somatic mutation carrier status (Martinez MF; Cells, 2019). This is also interesting to include when this particular issue is discussed.

: As suggested, the discrepancy in SMO expression in BCC between studies has been discussed and included in the revised manuscript (In page 8, line 20-25). 

“Interestingly, SMO transcription was not altered in BCCs, whereas Martinez et al. recently showed an increased mRNA expression of SMO in BCCs from 20 nevoid basal cell carcinoma syndrome (NBCCS) patients mainly caused by PTCH1 gene mutations (24). This discrepancy is probably due to the different study groups because our study only includes sporadic BCCs that have different genetic profiles from those of NBCCS patients even though both BCCs share a common signaling abnormality.”

The correlation analysis between HH-related molecules and stem cells markers results challenging to this reviewer:

. In result text (page 8, lines 2-5), the correlation scores and P values for GLI1 probably correspond to the GLI2 (see Figure 6A and 7). Please, check the data and correct both values and its interpretation.

: The error in the result text has been fixed as pointed out. 



. Why the correlation of SMO and SHH mRNA levels with LGR4, LGR5, LGR6, and LRIG1 are missing? This issue need to be solved to understand overall relationship between HH pathway and stem cells markers.

: The correlation of SHH mRNA with stem cell markers has been included in the Figure 7 and result and figure 7 legend has been fixed. However, the SMO mRNA levels are too low to analyze the correlation. 

Some bars in the graphs, such as the EP in Figure 2E and the GLI IHC of Figure 5B, have no dispersion. Check carefully the data presentation throughout the manuscript.

 : There is no error bar in the IHC scores of EP because all values are same as 3. 

The figure 6B are confusing about the methodology used: 

• . If the authors try to show the in situ hybridization of GLI1 and LRG5, no mention concerning GLI1 in methods were found.

• . If the authors try to show IHC of GLI1 and LRG5 proteins, the entire legend of figure 6 need to be rewritten to clearly state which variable is represented in each case.

: LGR5 staining was performed by RNA in situ hybridization because there is no reliable antibody to LGR5 for immunohistochemical anaylsis with FFPE specimens, whereas GLI1 expression was detected by IHC. As suggested, the legend of figure 6 was rewritten to clarify the methodology. 

“Fig. 6 Correlations of GLI1 expression with other stem cell markers in basal cell carcinomas. (A) Scatter plots with regression lines showing the correlations between GLI1 and stem cell marker mRNA expression. (B) Representative case of basal cell carcinoma expressing both LGR5 mRNA and GLI1 protein (RNA in situ hybridization for LGR5 and immunohistochemistry for GLI1).”

Would important to add a comment in Introduction about the reason/s to study the stem cell markers and its relationship with HH pathway. 

: As suggested, the reason to study the relationship between HH pathway and stem cell markers has been included in the Introduction. (Page4, line 2-5)

Page4, line 2-6: “Although the precise cellular origin of BCC has been controversial, recent studies have demonstrated that BCC-like tumors can arise from multiple hair follicle (HF) stem cell populations (15-17). Since several distinct stem cell markers of HF have been identified by lineage-tracing experiments, we also assessed the correlation of HH molecules with the established markers including LGR4, LGR5, LGR6, and LRIG1.”

The amount of samples studied are confusing. In the first paragraph of subjects section of Materials and methods stated “…152 formalin-fixed, paraffin-embedded (FFPE) human skin tissues (normal skin, n = 5; skin tumors, n = 147)…” but in the next sentence and in Tissue microarray construction declared 168 skin tumors. Is important to correct and uniform the number of samples studied. In addition, since the clinicopathological features of patients are not analyzed, its mention in the text is unnecessary. 

: The tumor number has been changed from 147 to 168 and the sentence of clinicopathological features has been removed. 

The number of samples of each subtype of BCC have to be included in figure 5B.

: As suggested, the number of samples have been included.

The final sentence of Discussion point out the relationship between HH and Wnt signaling in BCCs, but this affirmation are not supported by the results. Then, would more appropriate a suggestion of link between both pathways.

: As suggested, the sentence of “indicating a functional relationship between~” has been replaced by “suggesting a link between~”

Minor changes:

- Second paragraph of Introduction, line 2: “…12transmembrane receptor for SHH that has a regulatory effect on the pathway (8). in the absence of…” replace by “…12 transmembrane receptor for SHH that has a regulatory effect on the pathway (8). In the absence of…”

- In page 5, line 4 replace Anti-PTCH2 by anti-PCTH2.

- In page 6, last paragraph, include HDA samples in “SHH expression was slightly higher in several tumors including BCC, TE, EP, and SPA (Fig. 1E)”

- In page 7, close de parenthesis in “…PMC (n = 13)…”

- In page 8, line 5, replace PTC2 by PTCH2.

: All minor errors have been fixed in the revised manuscript.

---

## [Decision Letter · Decision Letter 1]

5 Nov 2019

PONE-D-19-24006R1

Expression profile of Sonic hedgehog signaling-related molecules in basal cell carcinoma

PLOS ONE

Dear Dr Jang,

Thank you for submitting your manuscript to PLOS ONE. After careful consideration, we feel that it has merit but does not fully meet PLOS ONE’s publication criteria as it currently stands. Therefore, we invite you to submit a revised version of the manuscript that addresses the points raised during the review process.

1.  Address the writing for page 8, lines 7-8 as specified in reviewer 1's attachment.

2.  Ensure that when a gene/RNA is being referred to that the name is in italics (e.g. for in RNA in situs).  Protein names are not italicized.

We would appreciate receiving your revised manuscript by Dec 20 2019 11:59PM. To enhance the reproducibility of your results, we recommend that if applicable you deposit your laboratory protocols in protocols.io, where a protocol can be assigned its own identifier (DOI) such that it can be cited independently in the future. For instructions see: http://journals.plos.org/plosone/s/submission-guidelines#loc-laboratory-protocols

We look forward to receiving your revised manuscript.

Kind regards,

Amanda Ewart Toland, Ph.D.

Academic Editor

PLOS ONE

Reviewers' comments:

Reviewer's Responses to Questions

**Comments to the Author**

1. If the authors have adequately addressed your comments raised in a previous round of review and you feel that this manuscript is now acceptable for publication, you may indicate that here to bypass the “Comments to the Author” section, enter your conflict of interest statement in the “Confidential to Editor” section, and submit your "Accept" recommendation.

Reviewer #1: All comments have been addressed

2. Is the manuscript technically sound, and do the data support the conclusions?

Reviewer #1: Yes

3. Has the statistical analysis been performed appropriately and rigorously? 

Reviewer #1: Yes

4. Have the authors made all data underlying the findings in their manuscript fully available?

Reviewer #1: Yes

5. Is the manuscript presented in an intelligible fashion and written in standard English?

Reviewer #1: Yes

6. Review Comments to the Author

Reviewer #1: The revised version of the manuscript has been carefully checked and the author’s responses are satisfactory and greatly improve the overall quality of the paper. In this version, only one issue has to be corrected (please, see the attached file).

7. PLOS authors have the option to publish the peer review history of their article (what does this mean?). If published, this will include your full peer review and any attached files.

Reviewer #1: No

---

## [Author Response · Author response to Decision Letter 1]

5 Nov 2019

We appreciate the careful evaluation of our manuscript and revised our manuscript to address the issue raised by reviewer and editor. 

1. The revised version of the manuscript has been carefully checked and the author’s responses are satisfactory and greatly improve the overall quality of the paper. In the view of this version, only one issue has to be corrected:

In page 8, lanes 7-8, “… GLI1 expression showed a strong positive correlation with LGR4 (r2=0.62, P < 0.001), LGR5 (r2=0.60, P < 0.001), and LGR6 7 (r2=0.21, P = 0.05), but not with LRIG1…” needs to be rewrite as “… GLI1 expression showed a strong positive correlation with LGR4 (r2=0.62, P < 0.001), LGR5 (r2=0.60, P < 0.001), but not with LGR6 (r2=0.21, P = 0.05) and LRIG1…” . As is stated in statistical analysis section, “A P-value < 0.05 was considered statistically significant”, both LGR6 and LRIG1 have to be considered not significant.

: As suggested, the sentence has been rewritten in the revised manuscript.

---

## [Editor Report · Decision Letter 2]

7 Nov 2019

Expression profile of Sonic hedgehog signaling-related molecules in basal cell carcinoma

PONE-D-19-24006R2

Dear Dr. Jang,

We are pleased to inform you that your manuscript has been judged scientifically suitable for publication and will be formally accepted for publication once it complies with all outstanding technical requirements.

With kind regards,

Amanda Ewart Toland, Ph.D.

Academic Editor

PLOS ONE
---

## [Editor Report · Acceptance letter]

12 Nov 2019

PONE-D-19-24006R2 

Expression profile of Sonic hedgehog signaling-related molecules in basal cell carcinoma 

Dear Dr. Jang:

I am pleased to inform you that your manuscript has been deemed suitable for publication in PLOS ONE. Congratulations! Your manuscript is now with our production department. 

With kind regards,

on behalf of

Dr. Amanda Ewart Toland 

Academic Editor

PLOS ONE